# Numerical Assessment of the Thermal Performance of Microchannels with Slip and Viscous Dissipation Effects

**DOI:** 10.3390/mi15111359

**Published:** 2024-11-08

**Authors:** Pamela Vocale, Gian Luca Morini

**Affiliations:** 1Department of Engineering and Architecture, University of Parma, 43121 Parma, Italy; 2Department of Industrial Engineering, Alma Mater Studiorum Università di Bologna, Viale Risorgimento 3, 40135 Bologna, Italy; gianluca.morini3@unibo.it

**Keywords:** slip flow, viscous heating, noncircular cross-sections, single gas flow

## Abstract

Microchannels are widely used across various industries, including pharmaceuticals and biochemistry, automotive and aerospace, energy production, and many others, although they were originally developed for the computing and electronics sectors. The performance of microchannels is strongly affected by factors such as rarefaction and viscous dissipation. In the present paper, a numerical analysis of the performance of microchannels featuring rectangular, trapezoidal and double-trapezoidal cross-sections in the slip flow regime is presented. The fully developed laminar forced convection of a Newtonian fluid with constant properties is considered. The non-dimensional forms of governing equations are solved by setting slip velocity and uniform heat flux as boundary conditions. Model accuracy was established using the available scientific literature. The numerical results indicated that viscous dissipation effects led to a decrease in the average Nusselt number across all the microchannels examined in this study. The degree of reduction is influenced by the cross-section, aspect ratio and Knudsen number. The highest reductions in the average Nusselt number values were observed under continuum flow conditions for all the microchannels investigated.

## 1. Introduction

Originally developed for integrated circuits, the use of microchannels has since expanded across various fields. Devices such as micropumps, microturbines, microvalves, microreactors and compact heat exchangers are now commonly based on this versatile technology.

The performance of microchannels has been analyzed by many researchers, as highlighted by Harris et al. [1] in a recent review, because the thermohydraulic properties of microchannels are essential prerequisites for their use in thermal applications. In particular, Su et al. [2] numerically investigated the performance of a hydrodynamically fully developed but thermally developing flow in rectangular microchannels under a constant wall temperature. Their results showed that at the channel inlet, the local Nusselt number is independent of cross-sectional geometry, but differences in the Nusselt number for varying aspect ratios increase along the flow direction. Moreover, they developed new correlations for the actual Nusselt number and thermal entrance length in rectangular channels.

Kewalramani et al. [3] presented an experimental and numerical investigation of the performance of trapezoidal microchannels under H1 (axially constant heat flux with circumferentially constant temperature) and H2 (constant axial and circumferential wall heat flux) boundary conditions. They found that the Nusselt and Poiseuille numbers increased with both the side angle and aspect ratio of the cross-section. Correlations for Poiseuille number, Nusselt number, and entrance length for trapezoidal channels with side angles between 30° and 60° and aspect ratios from 0.1 to 10 were developed.

The thermal analysis of gaseous flows in trapezoidal microchannels was performed by Sadasivam et al. [4], who also investigated the performance of double-trapezoidal (hexagonal) cross-sections under H1 and T thermal boundary conditions. Their findings revealed a strong dependence of the friction factor and Nusselt number on the aspect ratio and trapezoidal angle.

However, a key question that arises with miniaturization is the suitability of traditional scaling laws. As devices shrink to the microscale, the traditional assumptions of continuum fluid mechanics often become invalid, necessitating a deeper analysis of scaling effects, such as rarefaction, viscous dissipation, compressibility, and axial conduction [5].

Rarefaction occurs when the mean free path of gas molecules approaches the same scale as the characteristic dimensions of the microchannel. Viscous dissipation refers to the conversion of kinetic energy into thermal energy due to viscous forces within the fluid. In microchannels, this effect is pronounced due to the high surface-to-volume ratio, which leads to significant heating of the fluid.

The combined effects of rarefaction and viscous dissipation create a complex interplay that influences microchannel performance. Rarefaction can reduce frictional losses and modify flow characteristics, while viscous dissipation can lead to increased temperatures and affect fluid properties. Analyzing these effects requires an integrated approach that considers both fluid dynamics and thermal phenomena.

The effects of rarefaction and viscous heating on slip flows with uniform heat flux at the wall have been studied in microchannels featuring several cross-sections (i.e., circular, rectangular, elliptical, and rhombic cross-sections). In particular, the impact of viscous dissipation on circular microchannels was investigated by Tunc and Bayazitoglu [6], who considered only two Brinkman number values: *Br* = 0.01 and *Br* = −0.01. Their findings showed that the Nusselt number can be reduced by up to 8% for *Br* = 0.01. The investigation of the viscous dissipation influence on the convective heat transfer in microducts was extended by Aydın and Avcı [7] by considering more Brinkman numbers.

The effect of viscous dissipation on the Nusselt number in rectangular microchannels was studied by van Rij et al. [8]. Their results, presented for only two values of the Brinkman number (i.e., *Br* = 0.05 and *Br* = −0.05), highlighted that for *Br* = 0.05, the Nusselt number exhibited a maximum rate of change of approximately 24%.

Vocale et al. [9] studied the influence of viscous dissipation in elliptical microchannels by considering several values of the aspect ratio of the ellipse and several values of the Brinkman number. They found that the Nusselt number exhibited a maximum rate of change of approximately 30%.

Recently, Vocale et al. [10] studied the impact of viscous heating on rhombic microchannels. Their findings, presented for many values of the side angle and the Brinkman number, reveal that the Nusselt number exhibited a maximum rate of change of approximately 25%.

Viscous dissipation effects in trapezoidal microchannels characterized by an apex angle equal to 54.74° were analyzed by Kuddusi [11] by considering three sides of the cross-section heated, and one was considered adiabatic. Their findings, presented for only one value of the Brinkman number (i.e., *Br* = 0.005), reveal that the Nusselt number exhibited a maximum rate of change of approximately 2%. Moreover, they presented the rate of change in the Nusselt number for different values of the Brinkman number (i.e., up to 0.5), but only for an aspect ratio equal to 0.4.

Slip flows in double-trapezoidal (hexagonal) microchannels have been investigated by many researchers [12], who have also presented simplified slip models to evaluate friction factors in polygonal microchannels.

Recently, the thermal performance of slightly rarefied gas flowing through circular microchannels under constant wall temperature conditions was experimentally investigated by Hemadri et al. [13]. They found that the Nusselt number exhibits only a weak dependence on the Knudsen number within the investigated range (0.001 < *Kn* < 0.012).

Srinivasan et al. [14] presented an experimental and numerical investigation of the tangential momentum accommodation coefficient (TMAC) for smooth and rough microchannel surfaces within slip-to-transition regimes. By using nitrogen as a working fluid, they observed lower TMAC values compared to prior results for rough surfaces, concluding that decreasing surface roughness results in an increase in slip velocity. The impact of the accommodation coefficient on the performance of compressible and slightly rarefied nitrogen gas flow in microchannels was also numerically investigated by Mebarki et al. [15]. Their findings showed that decreasing the tangential momentum accommodation coefficient increased gas velocity and the velocity jump near the walls while reducing the pressure gradient.

The literature review clearly indicates that understanding the effects of rarefaction and viscous dissipation is critical for the design and optimization of microfluidic systems.

Furthermore, the review highlights that existing data for rectangular microchannels in the literature cover only a few values of aspect ratio, Knudsen number, and Brinkman number for rectangular microchannels, although in microchannel heat sink (MCHS) designs, rectangular- (62.5%) and square- (10%) shaped channels account for approximately three-quarters of all microchannel structures [1].

Finally, the literature review revealed that the analysis of slip flow in trapezoidal and double-trapezoidal (hexagonal) microchannels was investigated only in a few works, although these cross-sections are commonly used in various compact heat exchangers because they can be easily obtained by using chemical etching on silicon wafers. In particular, trapezoidal microchannels are used in electronic cooling systems [16], while double-trapezoidal microducts are widely used in chemical applications [17].

To fill this gap, the present analysis offers a numerical analysis of the performance of rectangular, trapezoidal and double-trapezoidal microchannels. This study considered a Newtonian fluid with constant properties in fully developed laminar forced convection. The governing equations were solved using a commercial partial differential equations (PDE) solver, with slip velocity and uniform heat flux along the wetted perimeter as boundary conditions. The model’s accuracy was validated against the existing scientific literature.

The uniform wall heat flux boundary condition, identified as the H2 boundary condition in the literature (see Shah and London [18]), was selected because it is particularly useful in the presence of a microchannel with an imposed heat flux when solid walls are made of materials with low thermal conductivity.

Due to the high density and electrical conductivity of traditional materials, alternatives such as high-thermal-conductivity polymers and ceramics are being considered. Harris et al. [1] highlighted that although the use of copper (Cu) microchannels has remained steady over time, polymer-based materials (such as PDMS and PMMA) have shown an upward trend in recent years. In particular, they reported a trend of material usage in microchannels, emphasizing that Cu, Si, and PDMS tend to be the most common materials, accounting for 22.9%, 20%, and 17.1% of the total microchannels fabricated, respectively.

Moreover, in such cases, it is important to carefully monitor the rise in wall temperature along the wetted perimeter, especially near the corners of the cross-section, where temperature spots occur because of stagnant fluids.

Therefore, the novelty of the proposed manuscript is that it investigates the impact of rarefaction and viscous dissipation on the performance of microchannel cross-sections commonly used in micro heat sinks and micro heat exchangers under a thermal boundary condition that characterizes the behavior of micro devices made of polymer-based materials. The basic assumptions employed are that the effect of compressibility is neglected, the flow is fully developed, and the fluid properties are constant. To apply the model adopted in the present study, it is necessary to verify the validity of the assumptions. However, the numerical model adopted here provides benchmark results for other approaches, such as approximate or numerical methods. From this perspective, the numerical analysis provided here represents a valuable resource for designers and engineers involved in the sizing of micro heat sinks and micro heat exchangers, as well as Knudsen pumps.

## 2. Analysis

### 2.1. Mathematical Model

This study focuses on analyzing fully developed laminar gas flow through parallel microchannels, such as micro heat sinks (see Figure 1). The gas moves through microchannels with rectangular, trapezoidal and double-trapezoidal (or hexagonal) cross-sections, as shown in Figure 2. The investigated microchannels were characterized by an aspect ratio *γ* = *d*/*w*. Moreover, the trapezoidal microchannels presented an apex angle (α in Figure 2) equal to 54.74°, which is typically obtained by using chemical etching on silicon wafers as a manufacturing technology [17,19]. A Cartesian coordinate system was employed, with the origin located at the center of the cross-section.

A two-dimensional analysis of the microchannel was conducted under the following assumptions:the flow was considered incompressible, laminar, fully developed, and in a steady-state condition;the fluid properties were assumed to be constant;the channel walls were considered rigid and non-porous;the effects of heat radiation were neglected.

Based on the above assumptions, the non-dimensional form of the Navier–Stokes equation can be written as follows:(1)$$\frac{\partial^{2}u^{*}}{{\partial x^{*}}^{2}}+\frac{\partial^{2}u^{*}}{{\partial y^{*}}^{2}}+p^{*}=0$$
where *u** and *p** indicate the dimensionless velocity and pressure gradients, respectively.

It must be highlighted that the gas can be assumed incompressible if the Mach number cannot be larger than 0.3 and the pressure drop cannot be higher than 10% of the initial static pressure [20].

Given the small temperature rise in the actual microchannel during thermal management, a constant-property assumption was used in this research. However, to validate this assumption regarding the thermophysical properties of the working fluid, the increase in its bulk temperature must be evaluated.

The validity of the assumption of a fully developed flow must be checked by evaluating the hydrodynamic and thermal entrance lengths. Estimations for these entrance lengths can be obtained using the correlations available in the literature [2,3,21].

The non-dimensional form of the energy equation can be written as follows:(2)$$\frac{\partial^{2}T^{*}}{{\partial x^{*}}^{2}}+\frac{\partial^{2}T^{*}}{\partial {y^{*}}^{2}}=\frac{u^{*}}{A^{*}}\left(P^{*}+Br\int_{A^{*}}\Phi^{*}dA^{*}\right)-Br\Phi^{*}$$
where *T** is the dimensionless fluid temperature, *A** and *P** represent the dimensionless cross-section area and perimeter, respectively, (i.e., *A** = *A*/*D_h_*^2^ and *P** = *P*/*D_h_*, being *D_h_* the microchannel hydraulic diameter), Φ* represents the viscous dissipation function in the non-dimensional form, and *Br* is the Brinkman number, which is defined as:(3)$$Br=\frac{\mu W^{2}}{qD_{h}}$$

The first-order slip flow boundary condition was applied to account for rarefaction effects in the evaluation of the velocity field [5]:(4)$$u^{*}-u_{w}^{*}=\beta_{v}Kn\left.{\frac{\partial u^{*}}{\partial n^{*}}}_{w}\right|$$
where *β_v_* is the velocity slip coefficient [8,22], and *Kn* is the Knudsen number, which is defined as the ratio of the mean free path over the hydraulic diameter.

The energy equation in the non-dimensional form was solved by considering the H2 boundary condition (i.e., uniform wall heat flux boundary condition [18]).
(5)$${\left.\frac{\partial T^{*}}{\partial n^{*}}\right|}_{w}=-1$$
where *n** is the non-dimensional vector normal to the wall.

The wall temperature was evaluated by considering the first-order temperature jump boundary condition [5]:(6)$$T_{w}^{*}=T^{*}+\beta_{t}Kn$$
where *β_t_* is the temperature jump coefficient [8,22].

It is interesting to observe how, under H2 boundary conditions, the wall temperature is only shifted to a constant value (which depends on the Knudsen number, the Prandtl number, the specific heat ratio, and the thermal accommodation coefficient) with respect to the fluid temperature near the wall.

It is important to note that only the fluid domain is considered in this numerical study, with thermal boundary conditions applied directly [3].

The average Nusselt number was calculated as follows:(7)$$Nu=\frac{hD_{h}}{\lambda }=\frac{1}{\bar{T_{w}^{*}}-T_{b}^{*}}$$
where *h* is the average convective heat transfer coefficient, *λ* is the fluid thermal conductivity, and $\bar{T_{w}^{*}}$ and *T*^*^*_b_* indicate the average wall temperature and fluid bulk temperature in non-dimensional forms, respectively.

The solution to Equations (1) and (2) can be obtained by using tools for solving nonlinear partial differential equations (PDEs), such as Matlab 2024b, Mathematica 14, and FlexPDE 8. In the present analysis, the governing equations were solved by using the equation-based modeling tools of COMSOL Multiphysics 6.2 through which specific models based on mathematical equations can be created. The solver relative tolerance was set to 1 × 10^−6^. Quadratic shape functions have been used to solve for both fluid velocity and temperature fields because of low flow velocities [23].

### 2.2. Mesh Independence Study

The mesh independence study was performed by considering three mesh refinements. For all generated meshes, the average Nusselt number was evaluated using Equation (6). Nusselt number variations with the number of mesh elements for the microchannels investigated here are presented in Figure 3, Figure 4 and Figure 5 for several Knudsen numbers. For the sake of brevity, the results of the mesh independence study are reported only for the smallest value of the aspect ratio considered here (i.e., *γ* = 0.1), which is the worst case (i.e., shallow microchannels) for testing numerical convergence. It can be observed that for a number of elements (*N*) higher than 500, Nusselt numbers are independent of mesh size for all the cross-sections analyzed. The same findings were also observed for higher values of the aspect ratio.

### 2.3. Model Accuracy

Model accuracy was established using the available scientific literature. In particular, the numerical results for square microchannels (i.e., rectangular microchannels with *γ =* 1) are compared in Figure 6 with the experimental data presented by Demsis et al. [24] in terms of the Poiseuille number (*f Re*) within the slip flow regime and the numerical results obtained by van Rij et al. [8] in terms of the Nusselt in the slip flow regime and with viscous dissipation.

The comparison with experimental data revealed good agreement for the Knudsen numbers lower than 0.01, while the difference between the numerical Poiseuille numbers (*f Re*) and the experimental ones becomes larger for the Knudsen numbers higher than 0.01. This difference was explained by the authors considering that the experiments were carried out by assuming a pressure ratio ranging from 0.88 to 0.95; consequently, the rarefaction effects were not uniform throughout the tube.

The comparison between the numerical results presented in this study and the numerical results obtained by van Rij et al. [8] showed good agreement. In particular, the maximum and minimum differences were −0.58% and 0.77%, respectively, as shown in Table 1, where the data for *Br* = 0.05 are presented (i.e., the highest value of the Brinkman number considered in [8]).

The numerical results for trapezoidal microchannels, in terms of the Poiseuille number (*f Re*), were compared with the experimental data obtained by Vijayalakshmi et al. [25] in the continuum flow regime (Table 2) and with the numerical data presented by Morini et al. [19] within the slip flow regime (Table 3).

The agreement between the data reported in [19,25] and the present results can be considered satisfactory. In particular, the maximum and minimum differences between the present results and the data reported in [19] were 0.67% and −0.07%, respectively, as shown in Table 3.

The numerical results for trapezoidal microchannels, in terms of the Nusselt number, were compared with the data presented by Shah and London [18] in the continuum flow regime and without viscous dissipation. The agreement between the data reported in [18] and the present results can be considered satisfactory. In particular, the maximum and minimum differences between the present results and the data reported in [18] were 0.18% and 1.16%, respectively, as shown in Table 4.

The numerical results for double-trapezoidal microchannels were also compared with the data presented by Morini et al. [19] and Shah and London [18]. Specifically, the results from [19] were expressed in terms of the Poiseuille number (*f Re*) in the slip flow regime, whereas the data from [18] were provided in terms of the Nusselt number in the continuum flow regime, excluding viscous dissipation. The agreement observed, as illustrated in Table 5 and Table 6, can be considered satisfactory. In particular, the maximum and minimum differences between the present results and the data reported in [19] were 0.04% and −0.09%, respectively, as shown in Table 5.

## 3. Results and Discussion

The numerical results presented in this section cover different degrees of rarefaction by varying the Knudsen number and different aspect ratios. Specifically, the Knudsen number ranged from 0 to 0.1, while the aspect ratio *γ* varied from 0.1 to 1 for rectangular microchannels, from 0.1 to 0.707 for trapezoidal microchannels, and from 0.1 to 1.414 for double-trapezoidal microchannels. For a silicon microchannel, the aspect ratio is limited to a maximum value of tg(*φ*)/2, which is approximately 0.707 when *φ* = 54.74°. Beyond this point, the channel’s cross-section degenerates into a triangular shape. The double-trapezoidal (or hexagonal) cross-sections considered in the present study were obtained from two trapezoidal microchannels, limiting the aspect ratio to a maximum value of 1.414 when *φ* = 54.74°. Beyond this threshold, the channel’s cross-section transforms into a rhombic shape.

Additionally, nitrogen was assumed to be the working gas (i.e., *Pr* = 0.7 and *k* = 1.4), with both the momentum and thermal accommodation coefficients set to 1 [5].

### 3.1. Effect of Viscous Dissipation on Wall Temperature

Because viscous dissipation raises the fluid temperature at the wall, it is crucial to assess variations in wall temperature. From a practical standpoint, monitoring wall temperature is essential, particularly in applications where preventing it from reaching critical levels is mandatory.

The variation of the dimensionless wall temperature for rectangular microchannels with *γ* = 0.1, which represents the worst case (i.e., shallow microchannels) for fluid flow and heat transfer performances, is presented in Figure 7 for two values of the Brinkman number.

It can be observed that the non-dimensional fluid temperature varies significantly along the perimeter of the microchannel under the H2 boundary condition. This trend can be explained by recalling the significance of the H2 boundary condition, which imposes a uniform wall heat flux along the entire channel perimeter [18].

Moreover, it is evident that the wall temperature rises with an increasing Brinkman number. Additionally, the wall temperature becomes less influenced by the rarefaction effect (*Kn*) when the viscous effects are stronger (i.e., at higher Br values).

This effect can be explained by observing that viscous dissipation (*Br*) and rarefaction (*Kn*) play opposing roles: an increase in the wall slip velocity leads to a decrease in the velocity gradient and lowers heat generation close to the walls due to viscous dissipation. In Figure 7, Figure 8 and Figure 9, the abscissa (*s*) corresponds to the length of the perimeter, starting from point *O* in Figure 2.

The same findings were also obtained for trapezoidal and double-trapezoidal or hexagonal microchannels, as shown in Figure 8 and Figure 9, where the variation of the dimensionless wall temperature for *γ* = 0.1 is presented.

### 3.2. Effect of Viscous Dissipation on Nusselt Number

The thermal performance of the rectangular microchannels, in terms of the Nusselt number, is presented in Figure 10 for several Knudsen number values and two Brinkman number values. The data in Figure 10 show that for all values of the aspect ratio and the Knudsen number examined in this study, the Nusselt number decreases as the Brinkman number increases. This is because viscous dissipation generates thermal energy primarily near the wall, raising the fluid temperature at the wall. This increases the temperature difference between the fluid bulk temperature and the average wall temperature, leading to a reduction in *Nu* (see Equation (6)).

The data in Figure 10 also reveal that the impact of viscous dissipation diminishes as *Kn* increases. For *γ* = 1 and *Kn* = 0, a *Br* of 0.1 results in a 25% decrease in *Nu*, whereas at *Kn* = 0.1, the same *Br* leads to only a 5.5% reduction in *Nu*. This reduced effect of *Br* on *Nu* at higher *Kn* values is attributed to lower velocity gradients due to the increased wall slip. The same findings were obtained by van Rij [8], who analyzed only a few values of the aspect ratio and Brinkman number.

Figure 11 presents the thermal performance of the trapezoidal microchannels in terms of the Nusselt number for several values of the Knudsen and Brinkman numbers. The data shown in Figure 11 reveal that the well-known reducing impact of rarefaction on the Nusselt number is also confirmed for trapezoidal microchannels. Similar to the trend observed in [11] for trapezoidal microchannels characterized by different values of the apex angle, the Nusselt number decreased with an increasing aspect ratio. The influence of viscous dissipation was reduced for high Brinkman number values, even for trapezoidal cross-sections.

Figure 12 shows the variation in the Nusselt number with rarefaction and aspect ratio for double-trapezoidal (or hexagonal) microchannels. The data presented in Figure 12 reveal that the Nusselt number decreases as the Knudsen number increases for the double-trapezoidal (or hexagonal) microchannel. The one-way influence of the Knudsen number on the Nusselt number does not hold for the aspect ratio. The Nusselt number increases with the aspect ratio when *γ* < 0.8, reaches a maximum around *γ* = 0.8 and then decreases as the aspect ratio increases for *γ* > 0.8 for all values of the Brinkman number analyzed in the present analysis.

Similar trends were observed by Kuddusi et al. [11], who analyzed the impact of the aspect ratio in trapezoidal microchannels by considering three sides of the cross-section as heated and one as adiabatic. They concluded that the observed trends in Nusselt number were due to the considered thermal boundary conditions.

A final consideration regarding the results presented relates to the assumptions made in this study about the velocity slip (*β_v_*) and temperature jump (*β_t_*) coefficients. All results presented here are based on the assumption that *β_v_* = *β_t_* = 1.

When *β_v_* = 1, gas molecules transfer all their tangential momentum to the wall, resulting in diffuse reflection—a characteristic of rough solid surfaces. For smoother walls, a value less than one is typically expected. The value of *β_t_* depends on the value of *β_v_* and it is also influenced by the type of gas and wall surface considered.

Under the H2 boundary condition, *β_t_* directly affects the wall temperature (see Equation (6)); when *β_t_* < 1, a reduction in wall temperature is expected, along with an increase in the Nusselt number for a given Knudsen number (*Kn*). This observation highlights that the variation in the Nusselt number with *Kn* and the Brinkman number (*Br*) is strongly dependent on the value of *β_t_*.

However, accurately determining the values of *β_v_* and *β_t_* in practical applications remains a challenge, as the experimental data in the literature are sometimes contradictory. Further testing with materials commonly used in microfluidics is needed to address this knowledge gap.

## 4. Conclusions

This study has presented a numerical analysis of the performance of microchannels with rectangular, trapezoidal and double-trapezoidal (or hexagonal) cross-sections. The numerical model used in this paper considers the working fluid to be incompressible and with constant properties. The interaction between gas and channel walls was modeled by considering a fully diffusive reflection of the gas molecules (i.e., rough wall surface) and a fully thermal accommodation (*β_t_* = 1).

The key conclusions drawn from the analysis are as follows:Viscous dissipation effects result in a decrease in the average Nusselt number for all the microchannels investigated here; the extent of the reduction depends on the cross-section, the aspect ratio, and the Knudsen number.The lowest values of the average Nusselt number occurred under continuum flow conditions for all the microchannels investigated here.The Nusselt number exhibits a maximum rate of change of approximately 25% for rectangular microchannels.The Nusselt number exhibits a maximum rate of change of approximately 18% for trapezoidal microchannels.The Nusselt number exhibits a maximum rate of change of approximately 28% for double-trapezoidal (or hexagonal) microchannels.

The results presented here enable microthermal device designers to account for the effects of non-uniform wall temperature distribution in microchannels made using materials with poor thermal conductivity. In these cases, the applied heat flux at the walls can cause significant variations in wall temperature, potentially leading to considerable thermal deformation of the channel walls. This is particularly true for shallow microchannels, where temperature hotspots may develop near corners where the fluid is stagnant. From this perspective, the numerical analysis provided here represents a valuable resource for designers and engineers involved in the sizing of micro heat sinks and micro heat exchangers, as well as Knudsen pumps, where temperature gradients are generated in microchannels to create a pressure head.

Future research will examine the performance of microchannels under the H1 boundary condition, where the wall heat transfer rate remains constant along the axial direction and the wall temperature is uniform around the perimeter. Practically, the H1 boundary condition can be applied to microchannels made from highly conductive materials, such as copper and aluminum—two of the most commonly used materials for microchannels—which help maintain a uniform wall temperature around the channel.

## Figures and Tables

**Figure 1 micromachines-15-01359-f001:**
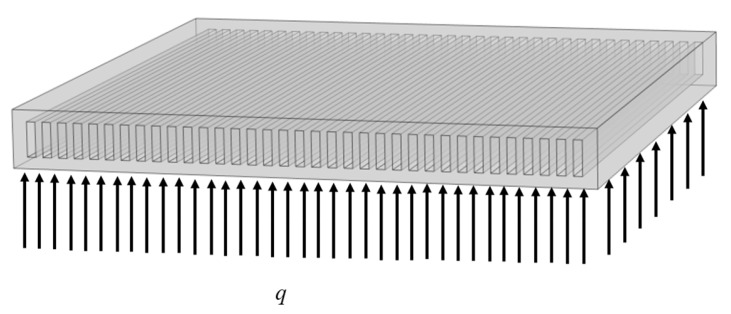
Schematic views of the parallel microchannels considered in the present analysis.

**Figure 2 micromachines-15-01359-f002:**
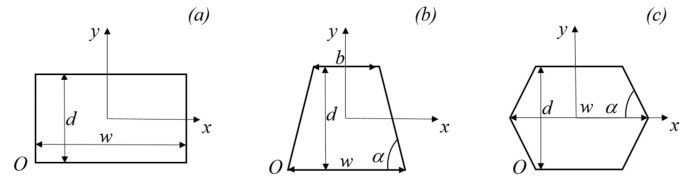
Microchannels under investigation: (**a**) rectangular; (**b**) trapezoidal; (**c**) double trapezoidal.

**Figure 3 micromachines-15-01359-f003:**
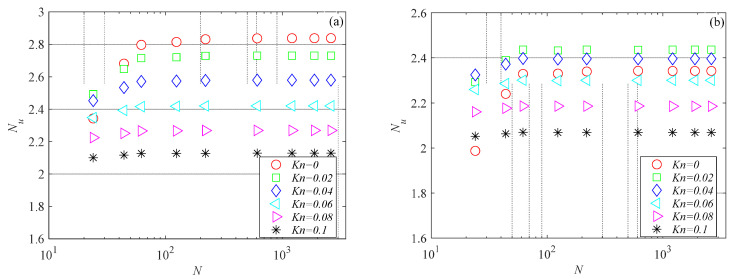
Nusselt number variation with number of mesh elements for rectangular microchannel with *γ* = 0.1: (**a**) *Br* = 0.01; (**b**) *Br* = 0.1.

**Figure 4 micromachines-15-01359-f004:**
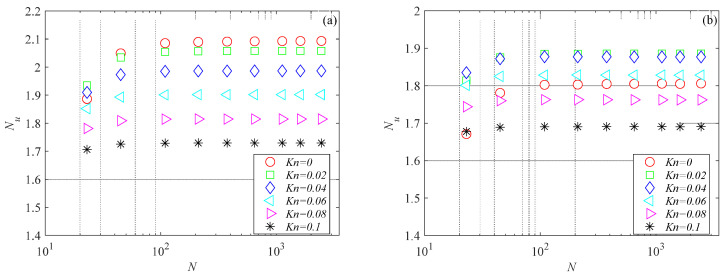
Nusselt number variation with number of mesh elements for trapezoidal microchannel with *γ* = 0.1: (**a**) *Br* = 0.01; (**b**) *Br* = 0.1.

**Figure 5 micromachines-15-01359-f005:**
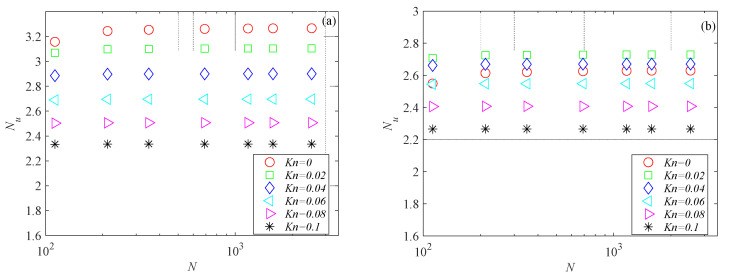
Nusselt number variation with number of mesh elements for double-trapezoidal microchannel with *γ* = 0.1: (**a**) *Br* = 0.01; (**b**) *Br* = 0.1.

**Figure 6 micromachines-15-01359-f006:**
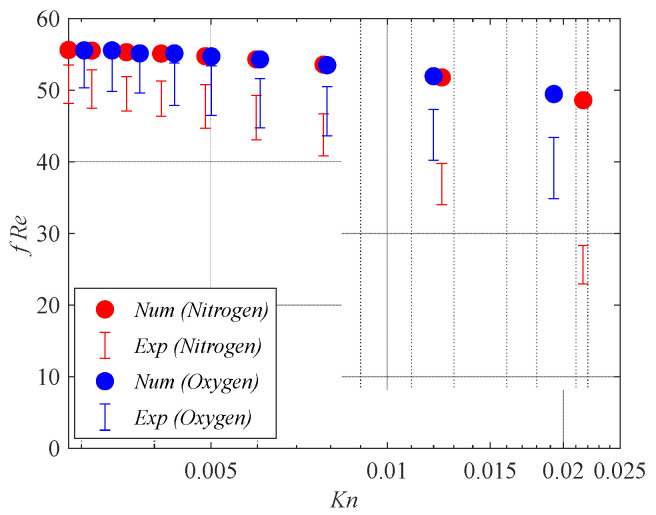
Comparison between the Poiseuille number (*f Re*) for square microchannels and the experimental data presented in [24] for Nitrogen and Oxygen gases (i.e., *γ* = 1 and *Br* = 0).

**Figure 7 micromachines-15-01359-f007:**
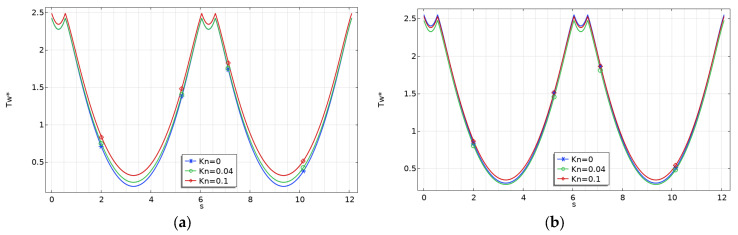
Variation of the dimensionless wall temperature for rectangular microchannels with *γ* = 0.1: (**a**) *Br* = 0; (**b**) *Br* = 0.1.

**Figure 8 micromachines-15-01359-f008:**
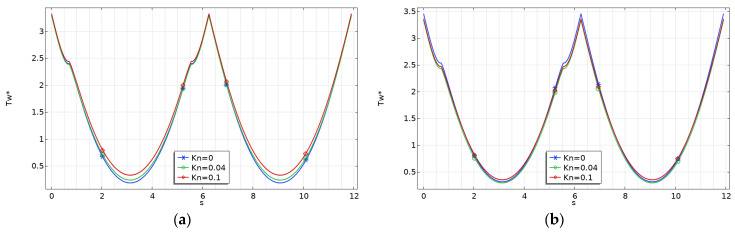
Variation of the dimensionless wall temperature for trapezoidal microchannels with *γ* = 0.1: (**a**) *Br* = 0; (**b**) *Br* = 0.1.

**Figure 9 micromachines-15-01359-f009:**
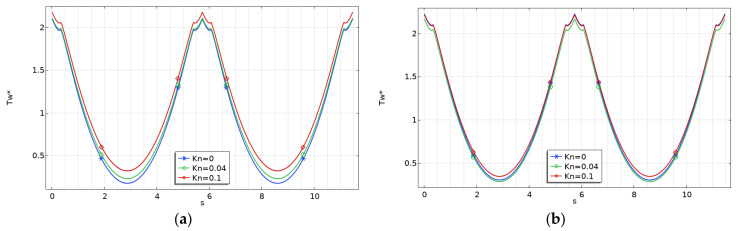
Variation of the dimensionless wall temperature for double-trapezoidal (or hexagonal) microchannels with *γ* = 0.1: (**a**) *Br* = 0; (**b**) *Br* = 0.1.

**Figure 10 micromachines-15-01359-f010:**
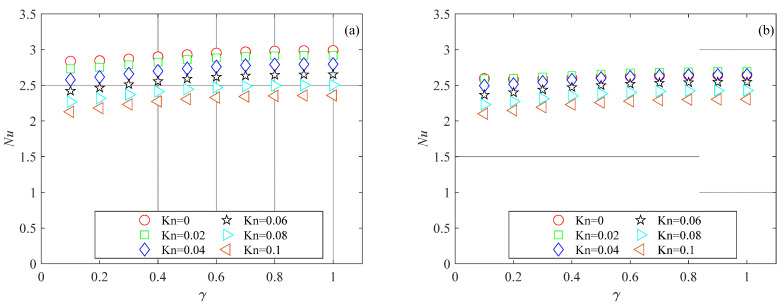
*Nu* as a function of the aspect ratio in rectangular microchannels for different values of the Knudsen numbers: (**a**) *Br* = 0.01; (**b**) *Br* = 0.1.

**Figure 11 micromachines-15-01359-f011:**
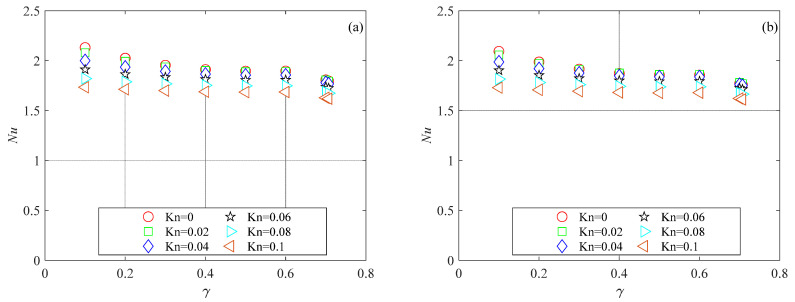
*Nu* as a function of the aspect ratio in trapezoidal microchannels for different values of Knudsen numbers: (**a**) *Br* = 0; (**b**) *Br* = 0.01; (**c**) *Br* = 0.05; (**d**) *Br* = 0.1.

**Figure 12 micromachines-15-01359-f012:**
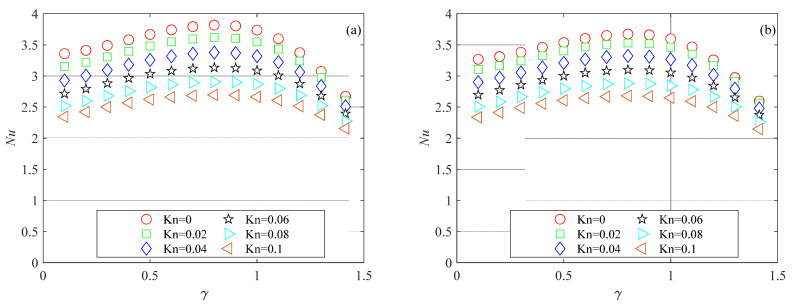
*Nu* as a function of the aspect ratio in double-trapezoidal (or hexagonal) microchannels for different values of the Knudsen numbers: (**a**) *Br* = 0; (**b**) *Br* = 0.01; (**c**) *Br* = 0.05; (**d**) *Br* = 0.1.

**Table 1 micromachines-15-01359-t001:** Comparison between Nusselt numbers for rectangular microchannels and the data presented in [8] for *Br* = 0.05.

*γ*	*Kn*	*Nu*	*Nu* [8]	Difference
0.2	0	2.600	2.585	−0.58%
1	0.08	2.410	2.429	0.77%

**Table 2 micromachines-15-01359-t002:** Comparison between the Poiseuille number (*f Re*) for trapezoidal microchannels and the experimental data presented in [25] (i.e., for α = 54.74°, and *Br* = *Kn* = 0).

*f Re*	*f Re* [25]	Difference
63.254	58.460	8.20%

**Table 3 micromachines-15-01359-t003:** Comparison between the Poiseuille number (*f Re*) for trapezoidal microchannels and the data presented in [19].

*γ*	*Kn*	*f Re*	*f Re* [19]	Difference
0.047	0.1	10.425	10.355	0.67%
0.62	0.1	7.663	7.669	−0.07%

**Table 4 micromachines-15-01359-t004:** Comparison between the Nusselt number for trapezoidal microchannels and data presented in [18] (i.e., for α = 60°, and *Br* = *Kn* = 0).

*γ*	*Nu*	*Nu* [12]	Difference
0.109	2.337	2.310	1.16%
0.782	2.094	2.090	0.18%

**Table 5 micromachines-15-01359-t005:** Comparison between the Poiseuille number (*f Re*) for double-trapezoidal microchannels and the results reported in [19].

*γ*	*Kn*	*f Re*	*f Re* [19]	Difference
0.8	0.1	8.463	8.460	0.04%
1.3	0.1	8.243	8.250	−0.09%

**Table 6 micromachines-15-01359-t006:** Comparison between the Nusselt number for double-trapezoidal microchannels and data presented in [18] (i.e., for α = 60°, and *Br* = *Kn* = 0).

*γ*	*Nu*	*Nu* [18]	Difference
0.866	3.8583	3.8620	−0.10%

## Data Availability

The raw data supporting the conclusions of this article will be made available by the authors on request.

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
