# Peer review of "Numerical Assessment of the Thermal Performance of Microchannels with Slip and Viscous Dissipation Effects"

_micromachines, 2024, doi:10.3390/mi15111359_

Round 1
Reviewer 1 Report
Comments and Suggestions for Authors
This manuscript reports a numerical analysis of the performance of microchannels with rectangular, trapezoidal, and double-trapezoidal cross-sections. This manuscript includes the viscous dissipation effects of the proposed microchannels. The manuscript is well-structured and organized. However, this manuscript must be improved based on the following issues:
1.- The abstract should include an introduction, the contribution, the main results, and a conclusion.
2.- The introduction should improve the description of the research problem. This section has various short paragraphs. The authors should include the novelty or scientific contribution of the proposed manuscript, considering the advantages and limitations compared to other research works reported in the literature. This section should incorporate recent and relevant references on the performance of microchannels.
3.- The authors should improve the description and schematic views of the models of proposed microchannels.
4.- What are the limitations of the proposed microchannel models?
5.- What are the future research works?
6.- The authors should include more discussions on the behavior of the results of Figures 4, 5, 6, 7, 8, and 9.
Comments on the Quality of English Language
Good
Reviewer 2 Report
Comments and Suggestions for Authors
The present paper aims to numerically simulate laminar, fully developed forced convection in microchannels with rectangular, trapezoidal, and double-trapezoidal cross-sections. Different aspect ratios are considered, as well as differnet values of the Knudsen number. The values of average Nusselt numbers are analyzed.
The following concerns should be addressed before the paper could be published in the Micromachines journal.
1. Please specify the full conditions of the problem under study. Is the gas considered to be incompressible? Why? It is well known that in microchannels, not only Mach number but also the ratio of the length of the microchannel to its hydraulic diameter are the parameters which define whether the gas is incompressible or not.
Is gas density also independent of the temperature?
2. In eq. 6, h means the average convective heat transfer coefficient, while earlier h was stated for the height of the microchannels. Please, use another letter.
3. Mesh independence study section. Please, redraw the plots in Figure 2 such that the points were not located right on the y-axis. There is double placement of letters a) and b) (at the left upper corners of the plot and beneath them). The y-axis limits are chosen unsuitably – there is no visible difference between point locations.
4. What is the goal of presenting Tables 2, 3, and 4? The mean and maximum deviation of the studied parameters should be presented instead.
5. In the conclusion section, it is written that the numerical model used is comparable to those employed by various authors. But the comparisons presented in Table 2, 3, and 4 are made for data obtained by one of the authors of the current manuscript. I would highly recommend making a comparison with experimental results (even for circular cross-sections) or numerical results of other authors.
6. Examining Figures 8-9, where dependencies of Nusselt numbers for different Knudsen numbers are presented, assures one that the largest Nu is for continuum flow (i.e. Kn = 0). While in the abstract section, there is a statement: “The lowest average Nusselt number values were observed under continuum flow conditions for all the microchannels investigated”. Please, explain.
Round 2
Reviewer 1 Report
Comments and Suggestions for Authors
The authors addressed the reviewer's comments. This version of the manuscript can be accepted for publication in Micromachines.
Comments on the Quality of English LanguageThe English grammar and style can be improved.
Author Response
The manuscript has undergone a thorough grammar check.
Reviewer 2 Report
Comments and Suggestions for Authors
The article contains multiple inaccuracies:
1. Please check all the references. For example, on page 5, " It must be highlighted that the gas can be assumed incompressible if the Mach number cannot be larger than 0.3 and the pressure drop cannot be higher than 10% of the initial static pressure [21]" should cite the work 23, not 21.
2. The numeration of the figures is incorrect. There are two figures numbered as Figure 2. The references to the figures should also be corrected.
3. Some sentences have a period in the middle of them, while others lack a period at the end.
Author Response
The authors thank the reviewer for this check.
The manuscript has been revised according to the Reviewer's comments.